# Therapeutic Approach Targeting Gut Microbiome in Gastrointestinal Infectious Diseases

**DOI:** 10.3390/ijms242115654

**Published:** 2023-10-27

**Authors:** Ziying Han, Yiyang Min, Ke Pang, Dong Wu

**Affiliations:** 1Department of Gastroenterology, State Key Laboratory of Complex Severe and Rare Diseases, Peking Union Medical College Hospital, Chinese Academy of Medical Sciences & Peking Union Medical College, Dongcheng District, Beijing 100730, China; 2Peking Union Medical College, Beijing 100730, China

**Keywords:** gastrointestinal infection, gut microbiome, fecal microbiota transplantation, probiotics, synthetic biology

## Abstract

While emerging evidence highlights the significance of gut microbiome in gastrointestinal infectious diseases, treatments like Fecal Microbiota Transplantation (FMT) and probiotics are gaining popularity, especially for diarrhea patients. However, the specific role of the gut microbiome in different gastrointestinal infectious diseases remains uncertain. There is no consensus on whether gut modulation therapy is universally effective for all such infections. In this comprehensive review, we examine recent developments of the gut microbiome’s involvement in several gastrointestinal infectious diseases, including infection of *Helicobacter pylori*, *Clostridium difficile*, *Vibrio cholerae*, enteric viruses, *Salmonella enterica* serovar Typhimurium, *Pseudomonas aeruginosa Staphylococcus aureus*, *Candida albicans*, and *Giardia duodenalis*. We have also incorporated information about fungi and engineered bacteria in gastrointestinal infectious diseases, aiming for a more comprehensive overview of the role of the gut microbiome. This review will provide insights into the pathogenic mechanisms of the gut microbiome while exploring the microbiome’s potential in the prevention, diagnosis, prediction, and treatment of gastrointestinal infections.

## 1. Introduction

The human gastrointestinal tract hosts trillions of microorganisms, including bacteria, fungi, archaea, protozoa, and viruses, the entire cluster of which is the human gut microbiota. The knowledge that microbes exist in the gut dates back centuries, but it is only in the last two decades that we have been able to depict the world of microbes inside our gastrointestinal tract and understand the immense influence that it has on the homeostasis and development of the host. The concept of gut microbiome not only includes the gut microbiota, which is a collection of living organisms, but also the metabolites of these organisms and environmental factors [1]. The gut microbiome is involved in many life activities, including the metabolism of dietary and endogenous substances, immune regulation, signal transduction to the central nervous system, and so on [2].

The impact of the gut microbiome on gastrointestinal infection has drawn special attention in recent years. It has been found that the gut microbiota plays an important role in maintaining homeostasis and resisting external disturbances. The concept of “colonization resistance” has been proposed to describe the ability of the host commensal microbiota to resist colonization of exogenous microbes with potential pathogenicity. This is modulated using the following mechanisms: competing for nutrients; secretion of factors that inhibit the growth of adjacent bacteria; enhancing epithelial barrier, either mucus or tight junction between epithelial cells; maturating the immune system, inducing and regulating immune reactions; and perhaps other mechanisms yet to be demonstrated [3,4]. Disruption of the gut microbiota, which may be due to dietary imbalance, use of antibiotics, proton pump inhibitors (PPIs), and some metabolism-associated drugs [3,5], may increase the susceptibility of the host to gastrointestinal infection. 

As such, the gut microbiome has been targeted as a new angle for viewing and dealing with gastrointestinal infectious diseases. Infection may alter the structure of the gut microbiome. These alterations may be diagnostic for the type of infection, useful in elucidating disease mechanisms and providing more specific therapies. Meanwhile, the gut microbiome appears as a novel and promising targeting point of therapy. In recent years, astonishing development has been made in therapies targeting the gut microbiome, including but not limited to probiotics, prebiotics, engineered probiotics, and fecal microbiota transplantation. However, controversy still exists. Not all research on the gut microbiota reached consistent conclusions, even about the same infectious disease. For example, some researchers claim that the gut microbiota promotes infection of enteric viruses [6,7], while others claim it is protective [8,9]. This review aims to summarize current evidence on the relationship between gut microbiome and several infectious diseases, which may differ considerably because of the different biological characters of the infectious agents. We selected these diseases based on their clinical significance and the quantity of existing research. We attempt to portray the influence that the diseases and the gut microbiome bring on each other while highlighting recent advances in how therapies can target the gut microbiome to treat these diseases.

## 2. *Helicobacter pylori*

### 2.1. Interaction of Gut Microbiome and Helicobacter pylori

*Helicobacter pylori* is a pathogen that colonizes the stomach of humans and is distributed worldwide. It was discovered as one major cause of chronic gastritis, peptic ulcer, and gastric cancer. Moreover, it has been found to be associated with many extra-gastric diseases, such as iron deficiency anemia and immune thrombocytopenic purpura [10].

*H. pylori* affects the host gastrointestinal environment using various mechanisms. By producing urease and multiple virulence factors such as vacuolating cytotoxin A (VacA) and high-temperature requirement A (HtrA), *H. pylori* disrupt the host cell junctions and epithelial barrier, damages host cells, and suppresses the host immune system [11,12]. *H. pylori* brings about significant changes to the gastrointestinal environment (for example, an increase in pH) and predictably influences the gut microbiota extensively [13].

A large amount of studies have attempted to characterize gut microbiota change resulting from *H. pylori* infection in humans. The results are not entirely consistent. A number of studies reported that *H. pylori* infection was associated with increased microbial α-diversity in and downstream the gastric environment, evaluated using the Shannon index, amplicon sequence variant (ASV) count, and other indicators [14,15,16,17]. Meanwhile, some other studies have found *H. pylori* infection to be negatively associated with gut microbiota diversity [18,19,20]. Since most studies excluded patients taking antibiotics and PPIs at least one month before sample collection, medications were not considered a major confounding factor in these studies. However, it should not be neglected that they may still be able to influence the results of gut microbiota analyses. 

In the context of microbiome composition changes, firstly and not surprisingly, in the gastric microbiome of *H. pylori* infectious patients, *H. pylori* becomes the dominant bacterium [19]. However, after the disease progresses into gastric carcinoma, it seems that the environment would no longer be suitable for *H. pylori* colonization, and *H. pylori*‘s relative abundance would decrease [21]. The specific changes in phyla and genera of microorganisms were different among studies, probably due to huge heterogeneity between study populations and patient status. The general trend included an increase in *Proteobacteria* and a decrease in *Bacteroidetes* [22,23,24,25]. In mouse models, Ralser et al. observed that *H. pylori* infection shifts the intestinal microbiota to a mucous-degrading direction, promoting the proliferation of *Akkermansia* and *Ruminococcus* [26]. Furthermore, using Mongolian gerbil models, Noto et al. discovered that the change of gastric mucosal microbiota structure brought by *H. pylori* infection was dependent upon the *H. pylori*-produced oncoprotein CagA [27].

Human gut fungi may also play a role in interacting with *H. pylori*. *Candida*, a common genus of fungi in the human microbiota, was known to be able to harbor *H. pylori* [28]. Currently, relatively little is known about how *H. pylori* infection changes gut mycobiota composition. Dash et al. found that the relative abundance of *Candida glabrata* and some other unclassified fungi was increased with *H. pylori* infection (Table 1) [16]. Other findings included the absence of *Agaricomycetes* in the proximal duodenum [17]. Additionally, antibiotic and PPI therapy for *H. pylori* might induce the flourishing of the fungal portion of the gut microbiota [29,30]. However, the detailed influence of *H. pylori* on gut fungal microbiota awaits further analysis.

### 2.2. Gut Microbiota Modulation for Therapy and Prevention 

Decades of efforts have been spent targeting *H. pylori* eradication. While successful eradication can improve patients’ health and may restore normal gut microbiota [37,38], traditional eradication therapies, usually using a combination of antibiotics (such as clarithromycin and amoxicillin) and PPIs, cannot always achieve ideal effects due to increasing antibiotic resistance and disturbance the drugs themselves bring on the gut microbiota [39,40]. Novel therapeutic approaches are needed, and probiotics present themselves as a promising choice.

Table 2 is a list of randomized controlled trials on the probiotic application in human *H. pylori* treatment published in the past 5 years. Overall, probiotics appear to be effective in relieving adverse symptoms, especially diarrhea. Results regarding their effect on the eradication rate are controversial. Only a part of the studies focused on microbiota change brought about using probiotics. Though there is evidence that probiotics can help build a healthier microbiota profile, more detailed research into the probiotic-induced microbiota change and its influence on host metabolism is still needed. Comparatively, *Lactobacillus* and *Saccharomyces* have been most extensively studied, and they likely represent the most potential probiotics in anti-*H. pylori* treatment, though the most effective probiotic, cannot yet be determined. Noticeably, cost-effectiveness studies on probiotic treatment are few.

Knowledge about the mechanism of the *H. pylori* inhibitory effect of probiotics remains limited. Relevant studies are mostly conducted in vitro. Boyanova et al. discovered that some *Lactobacillus* strains produced bacteriocin-like substances that exhibit strong inhibitory effects on *H. pylori* [41]. Studies on the immunomodulatory effects of *Lactobacillus* demonstrated that some strains could modulate immune reactions and alleviate inflammation by reducing the production of several pro-inflammatory cytokines, notably IL-8 [42], and stimulating the production of immunoregulatory cytokine IL-10 [43]. Do et al. proposed that *Lactobacillus rhamnosus* strain JB3 could inhibit the attack of *H. pylori* on host cells by mitigating lipid raft formation, which might be hijacked by *H. pylori* during infection [44]. One strain of *Parabacteroides golsteinii* was reported to have the ability to modify gut microbiota and reduce cholesterol levels, which helps reduce *H. pylori* toxicity [45].

Interestingly, probiotics have also been proposed as a preventive method against *H. pylori* infection. A study on *Limosilactobacillus fermentum* showed positive results, though studies in this direction are currently small-scale and lack long-term follow-up [46].

In general, evidence for the effectiveness of probiotics in treating human *H. pylori* infection is quite strong. In the future, it might be meaningful to investigate the cost-effectiveness of different probiotics in order to comprehensively assess their value in *H. pylori* treatment. 

**Table 2 ijms-24-15654-t002:** Randomized clinical trials on probiotic treatment for *Helicobacter pylori* infection.

**Probiotic Genus**	**Specific Probiotics Used**	Author	Year	**Patient Status before Enrollment**	**Combination with Traditional Therapy ***	**Effect**
** *Bacillus* **	*B. clausii*	Plomer [47]	2020	Untreated (in the previous 3 months)	Yes	Incidence and duration of diarrhea ↓ (*p* < 0.05)
** *Bifidobacterium* **	tetragenous	He [48]	2022	Untreated	Yes	Incidence of adverse events ↓ (*p* = 0.016) fluctuations of gastric microbiota ↓
** *Clostridium* **	*C. butyricum*	Chen [23]	2018	Untreated	Yes	Relieves more symptoms
***Lactobacillus* (including *Lacticaseibacillus*, *Lactiplantibacillus* **)**	*Lactobacillus acidophilus* and *L. rhamnosus*	Chen [49]	2021	Untreated	No	*H. pylori* bacterial load (measured using delta over baseline value, DOB) ↓ (*p* = 0.045)
	*L. crispatus*/*L. helveticus*/*L. plantarum*	Wang [50]	2022	Untreated (in the previous 1 month)	No	Eradication rate ↑ (especially *L. crispatus*) (*p* = 0.0039) Gastrointestinal symptoms ↓ (*p* < 0.001)
	*L. reuteri* (non-viable)	Yang [51]	2021	Untreated	Yes	frequencies of abdominal distention (*p* = 0.01) and diarrhea ↓ (*p* = 0.022) GSRS score ↓ (*p* = 0.03) More beneficial microbiota profile
	*Lacticaseibacillus paracasei* and *L. rhamnosus*	Guillemard [29]	2021	Untreated	Yes	SCFA ↑ (*p* = 0.035) Faster recovery of gut microbiota
	*L. reuteri*	Dore [52]	2022	Untreated (in the previous 1 month)	Yes	Cannot restore gut microbiota
** *Saccharomyces* **	*S. boulardii*	Qu [53]	2022	Treated and failed	No	Eradication rate ↑ during the first phase of treatment (*p* < 0.001) cost-effectiveness ratio ↑
	*S. boulardii*	Seddik [54]	2019	Untreated (in the previous 4 weeks)	Yes	Eradication rate ↑ (*p* = 0.02) Adverse events ↓ (*p* < 0.001)
	*S. boulardii*	Zhao [55]	2021	Untreated	Yes	Incidence of severe diarrhea ↓ (*p* = 0.04) Duration of diarrhea ↓ (*p* = 0.032)
	*S. boulardii*	Chang [56]	2020	Untreated (in the previous 1 month)	Yes	No effect on eradication rate and adverse reactions
**Mixed**	*L. plantarum* and *Pediococcus acidilactici*	McNicholl [57]	2018	Treated	No	No difference in eradication rate, side effects, and compliance
	*Bacillus subtilis* and *Enterococcus faecium*	Tang [39]	2021	Untreated	Yes	More beneficial microbiota profile
	*L. acidophilus*, *Lactiplantibacillus. plantarum*, *Bifidobacterium lactis*, *S. boulardii*	Viazis [58]	2022	Untreated	Yes	Eradication rate ↑ (*p* = 0.028) Side effects ↓ (*p* < 0.00001)

* Traditional therapy indicates therapies that consist mainly of antibiotics, proton pump inhibitors, and possibly bismuth. Therapies taken in the above research are majorly triple therapy and bismuth quadruple therapy. ** 23 new genera were separated from the original *Lactobacillus* genus, including *Lacticaseibacillus*, *Lactiplantibacillus*, and others. *Lacticaseibacillus rhamnosus* was formerly known as *Lactobacillus rhamnosus*. *Lactiplantibacillus* was formerly known as *Lactobacillus plantarum* [59].

## 3. *Clostridium difficile*

### 3.1. Interaction of Gut Microbiome and Clostridium difficile

*Clostridium difficile* infection (CDI) has been a common cause of healthcare-associated diarrhea, and 20–30% of patients suffered CDI recurrence, leading to a substantial health care burden [60,61]. *C. difficile* can colonize in the gut of healthy individuals, especially in infants, where colonization rates can reach 70% [62,63]. The gut microbiota of the healthy population might resist the infection of *C. difficile* [64]. Probiotics colonized in the gut, such as *Bifidobacterium longum*, showed a negative correlation to *C. difficile* colonization [62,65]. Primary and secondary bile salts, regulated by gut microbiota, were involved in the germination of *C. difficile* and enhanced the resistance of CDI [66,67]. Sterol metabolites of fecal samples could help distinguish CDI from healthy population [68].

CDI was related to gut microbiota dysbiosis. Antibiotic use, increasing age, and the use of PPIs are linked to a higher risk of CDI, potentially attributed to the disruption of gut microbiota homeostasis in patients. These factors have also been found to be associated with lower levels of colonized probiotics [61]. Antibiotic use could alter the microbiota profile, producing a suitable microenvironment for CDI [69].

Several studies have evaluated the gut dysbiosis of patients with CDI compared to healthy individuals. CDI patients showed a significant reduction in microbiota diversity compared to healthy individuals [64,70]. Manges et al. found an increase in *Firmicutes*, *Proteobacteria*, and *Actinobacteria* phylum, as well as a decrease in *Bacteroidetes* in CDI patients [71]. The change of *Bacteroidetes* and *Firmicutes* were identified as two independent risk factors of CDI [71]. A similar tendency of gut microbiota alterations was found in other studies [70,72,73,74]. Butyrate-producing bacteria, such as *Bifidobacterium* and *Bacteroides*, were negatively related to the abundance of *C. difficile* [70,73,75]. However, the abundance of *Enterococcus* increased significantly in CDI patients compared to healthy adults and was positively correlated with a worse prognosis [73]. *Enterococci* were linked to higher inflammation status and could enhance the pathogenesis of CDI by enhancing toxin production of *C. difficile* and reshaping metabolome, while the elimination of *Enterococci* leads to the delayed colonization of *C. difficile* [76]. Gomez-Simmonds et al. conducted long-term detecting of the gut microbiota in transplant patients, confirming microbiome dysbiosis as a risk factor for CDI [77]. A model combining gut microbiome and immune markers effectively distinguished CDI patients from asymptomatic carriers and patients with other causes of diarrhea [74].

Changes in fungal colonization were relatively ignored, but growing evidence revealed that fungi alterations were also implicated in CDI pathogenesis (Table 2). Compared to non-CDI diarrhea, CDI patients exhibit a higher abundance of *Penicillium* [31,33]. Compared to healthy controls, CDI patients have lower fungal diversity with enrichment of *Candida albicans*, which tends to increase after antibiotic treatment [32]. *C. albicans* has been implicated in providing a suitable environment for *C. difficile*, and oral supplementation of *C. albicans* is associated with an increase in disease severity and worse prognosis in CDI [78,79]. However, the study also found that pre-inoculation of *C. albicans* might enhance host resistance to CDI [80]. The specific role of fungi in CDI still needs further studies. 

### 3.2. Gut Microbiota Modulation for Therapy and Prevention 

Considering the potential involvement of gut microbiota in CDI, restoring gut homeostasis is crucial for CDI recovery. Gut modulation therapies, including gut metabolites, probiotics, and FMT, have been applied to manage CDI, especially recurrent CDI.

FMT has been included in medical guidelines, serving not only as therapeutic interventions but also as preventive measures against CDI. FMT was safe and had a high-efficiency rate for CDI ranging from 80% to 94%, especially in recurrent CDI [60,81,82]. More than 90% of CDI patients who responded to FMT resisted CDI recurrence [81,83]. A meta-analysis analyzed thirty-seven clinical trials and confirmed that the efficiency of FMT in recurrent CDI surpassed vancomycin and other antibiotics [84,85]. The efficacy of FMT is associated with the frequency and infusion number but not associated with whether it is fresh or frozen [60,84]. The efficacy of FMT in CDI patients is tightly correlated with the gut microbiome [32]. Before treatment, elevated levels of *Candida* in CDI patients were linked to a less favorable response to FMT, with the non-responsive FMT group exhibiting a higher *Candida* abundance compared to the responsive group [32]. FMT could modulate gut microbiome and increase gut diversity [68]. After FMT treatment, the composition of gut microbiota in the responder was similar to the FMT product, and the similarity lasted more than 1 year, while the non-responder showed little convergence [81]. An FMT product, “Rebyota”, is currently available for recurrent CDI treatment and has been approved by the Food and Drug Administration (FDA) in 2022 [86]. Rebyota is also expected to reduce medical expenses for recurrent CDI patients [87].

Probiotics have been widely applied in the management of CDI, but the effectiveness of probiotics is still controversial. In a randomized controlled trial (n = 2810), no conclusive evidence of probiotics’ preventive effects on CDI was observed [88]. However, in this study, the risk of CDI was only 1.2% [88]. A comprehensive meta-analysis incorporating thirty-one randomized trials (n = 8627) indicated that probiotics have the potential to effectively reduce the risk of CDI and ameliorate CDI symptoms without introducing an elevated risk of adverse effects [89]. The effectiveness of probiotics against CDI has been observed regardless of patient age, types of probiotic strains, and dosages of probiotics [89]. The supplement of *Bifidobacterium* could decrease the severity of CDI as well as the inflammatory status [78]. Tube feeding of *Lactobacillus casei* was safe and could prevent CDI in ICU patients [90]. Louie et al. revealed that probiotics reduced the recurrence rate of CDI [91]. However, there was no significant difference in *C. difficile* colonization after probiotics treatment [89]. Some studies suggest that probiotics could not effectively establish gut microbiota homeostasis, resulting in the lack of intestinal resistance to CDI [61].

Although the evidence was insufficient, emerging methods of gut microbiota modulation may offer the potential to control CDI. The level of short-chain fatty acids (SCFA) in the gut decreases after antibiotic treatment [69]. Higher SCFA levels in the gut can reduce the infection ability of *C. difficile* [92]. Among these SCFAs, butyrate shows an outstanding ability to protect CDI in multiple animal models [69,93]. CDI patients have a reduced abundance of butyrate-producing bacteria, resulting in a lower level of butyrate [70,92]. Both oral supplements of butyrate and butyrate-producing bacteria can effectively restrain *C. difficile*-induced colitis. This occurs via a multi-pronged mechanism involving the regulation of immune cells, the modulation of HIF pathways, and adjustments in bile acid metabolism [94,95,96,97]. Pensinger et al. even observed that butyrate could inhibit almost all *C. difficile* strains and decrease *C. difficile* burden [92]. However, Fachi et al. argued that butyrate exerts its protective effect by regulating the intestinal mucosal barrier rather than directly inhibiting the growth or toxin production of *C. difficile* [96]. Although multiple studies emphasize the efficacy of butyrate, its application in CDI requires further investigation. 

Emerging techniques, such as synthetic biology, have guided a new direction for CDI therapy. Engineered probiotics or products could enhance the gut barrier, precisely neutralize the toxin of *C. difficile*, increase gut resistance to CDI, and are expected to be a new potential drug for CDI in the future [98,99,100]. In conclusion, the crucial part of CDI therapy is to establish gut homeostasis to enhance resistance against *C. difficile*, involving the suppression of pathogenic bacterial growth and the promotion of probiotics. 

In conclusion, FMT has robust evidence in treating CDI, while the effectiveness of probiotics and SCFA are still controversial. Nevertheless, synthetic biology approaches may enhance probiotic applications.

## 4. Cholera

### 4.1. Interaction of Gut Microbiome and Cholera

Cholera, caused by *Vibrio cholerae* (*V. cholerae*) O1 or O139 bacteria, is an acute watery diarrheal disease that can cause severe dehydration and death [101]. Cholera, which holds the position as the third leading cause of diarrheal mortality, remains a prevailing issue in regions like Bangladesh and West Bengal, contributing to over 2.86 million infections and over 95,000 fatalities globally [102]. Rehydration therapy and antibiotics are the most common treatments. The increasing antibiotic resistance has prompted physicians to search for new therapy.

Cholera toxin was thought to be the main mechanism of pathogenesis of *V. cholerae*. However, recent studies found that an interbacterial killing device in *V. cholerae*, called type VI secretion system, could increase the virulence of *V. cholerae* by attacking gut microbiota and promoting epithelial shedding [103,104]. Quorum Sensing and Reactive Oxygen Species pathways allow *V. cholerae* and the gut microbiota to interact [105]. These findings indicated that gut microbiota might participate in the pathogenesis of cholera.

Due to disease prevalence and ethical concerns, current studies about gut microbiota in cholera primarily rely on animal models. Human population studies in this regard often suffer from limited sample sizes. The gut microbiota in cholera-endemic areas were specific, which was dominated by *Streptococci*, *Enterococcus faecalis*, and *Escherichia coli* [106]. *V. cholerae* could adapt to the environment by adjusting gene expression. Multiple studies have found gut dysbiosis and reduced microbiota diversity in cholera patients. Previous studies have demonstrated that during the acute infection phase, *V. cholerae* accounts for more than half of the gut microbiota composition in cholera patients [107]. Moreover, *V. cholerae* was involved in gut barrier damage, and the interaction between *V. cholerae* and gut microbiota inhibits gut reparations [103]. Healthy gut microbiota could decrease the virulence and protect against the colonization and infection of *V. cholerae*, while microbiota dysbiosis could promote the infection. Higher abundance of *Blautia obeuma* and *Ruminococcus obeum* were both negatively related to *V. cholerae* colonization [106,107]. *Blautia obeuma* could downregulate the virulence gene of *V. cholerae* by adjusting bile acid pathways [106]. *Ruminococcus obeum* was significantly increased after incubation of *V. cholerae* in mice. Supplement of *Ruminococcus obeum* could restrict the burden of *V. cholerae* [107]. Metabolite produced by gut microbiota also plays an important role in the resistance of *V. cholerae*. Bioactive compounds generated by gut microbiota impede *V. cholerae* motility and its ability to penetrate mucin [108]. Colibactin, a genotoxin generated by *Escherichia coli*, could inhibit the colonization of *V. cholerae* and is associated with the better prognosis of cholera patients [109]. In addition, studies about the fungal change of cholera infection are still lacking. 

### 4.2. Microbiota in Disease Diagnosis and Prediction

Microbiota dysbiosis could help the diagnosis and prediction of cholera. Mao et al. designed a *Lactococcus lactis* biosensor that could detect the presence or infection of *V. cholerae* by detecting the Quorum Sensing in *V. cholerae* [110]. This living diagnosis approach has the potential to significantly reduce turnaround time, lower testing costs, and facilitate early intervention. Moreover, engineered bacteria have great potential in monitoring cholera spread, even in asymptomatic *V. cholerae* carriers.

*V. cholerae* has been detected in the intestines of healthy individuals in cholera-endemic areas, but only a fraction of this population contracts cholera with symptoms [111]. Therefore, early identification of individuals with high risk of cholera is important. Among close contacts of cholera patients, the *Bacteroidetes* phylum was tightly related to cholera susceptibility, while *Paracoccus aminovorans* facilitates *V. cholerae* colonization [112]. The composition of gut microbiota in these close contacts has the potential to predict cholera occurrence [112]. Levade et al. confirmed the prediction ability of gut microbiota and highlighted that metagenomic sequencing could improve efficacy and accuracy [113]. Despite its slightly lower precision, gut microbiota can also assist in predicting disease severity in cholera contacts [113]. 

### 4.3. Gut Microbiota Modulation for Therapy and Prevention 

Limited research has been conducted on gut modulation therapy in cholera patients, with the majority of studies relying on animal models. A recent study revealed that SCFA could stimulate the immune response to cholera toxin via G-protein-coupled receptor 43 (GPR43) [114]. SCFA has demonstrated the ability to inhibit dehydration induced by cholera toxin, especially butyrate [115]. Conversely, rehydration could also elevate the level of SCFA within the gut [116]. However, there is no evidence about the efficacy of SCFA for cholera patients.

Although several studies found the therapeutic potential of probiotics and FMT in animal cholera models, the function of probiotics and FMT in cholera patients remains unknown. Jayaraman et al. designed and engineered *Escherichia coli* with a sense-and-kill system to precisely kill *V. cholerae* [117]. Mao et al. found that supplements of engineered probiotics could decrease the bacteria burden of *V. cholerae*, inhibit infection, and decrease mortality in mouse models [110]. Engineered probiotics showed the potential to prevent and treat cholera [110]. Mice that received gut microbiota from healthy individuals showed resistance to *V. cholerae* colonization. However, significant variations were observed in the ability of fecal samples from different donors to inhibit *V. cholerae* colonization [106]. 

Several studies have indicated the potential involvement of probiotics in immune responses to vaccines. It has been observed that undernourished children typically display a weaker reaction to oral cholera vaccines, potentially linked to their disrupted gut microbiota. Supplementation with prebiotics and probiotics could improve the intestinal IgA immune response in mice colonized with the gut microbiota of undernourished children [118]. However, two randomized controlled trials found that supplements of *Bifidobacterium breve* did not enhance the host immune response to the cholera vaccine but were related to increased immunoglobulin and lower levels of *Enterobacteriaceae* [118,119]. 

A new application of synthetic biology in cholera is the probiotics cholera vaccine. Using nine gene modifications in the Haitian outbreak strain, a live cholera vaccine was constructed and has probiotic properties. The live cholera vaccines could inhibit the colonization of *V. cholerae* before adaptive immunity was stimulated [120]. Further research revealed that this live cholera vaccine not only provides cholera prevention in various animal models but also enhances offspring’s resistance to cholera [121]. These suggested that engineered vaccines combine the benefits of vaccines and probiotics, not only stimulating intestinal responses but also facilitating the establishment of long-term immunity. Probiotics could serve as a new cholera prevention strategy in the future.

In conclusion, the gut microbiota holds significant potential for the diagnosis and prediction of cholera. Particularly, synthetic biology techniques have opened new avenues for the diagnosis and treatment of cholera.

## 5. Enteric Viruses

### 5.1. Interaction of Gut Microbiome and Enteric Viruses

Viral gastroenteritis has become a significant public health problem. Among the enteric viruses, there are over 100 identified serotypes. Rotavirus and norovirus are two of the most prevalent enteric viruses [122]. Studies have found that the virus could directly bind to commensal bacteria and fungi, such as *Enterobacter cloacae* and *Candida albicans* [9,123]. Therefore, the gut microbiota and gut immune system might play a role in the pathogenesis of viral infection.

However, the role of gut microbiota in viral infections seems to be controversial. Some studies agreed that gut microbiota may promote viral infection. For instance, in rotavirus infection, germ-free and antibiotic-treated mice exhibited reduced viral replication, delayed viral shedding, extended immune response, and milder symptoms [6]. Similar trends were also observed in norovirus infection [7]. A disrupted microbiota was associated with a higher level of lipopolysaccharide (LPS), which could increase viral stabilization [124]. Mechanistically, the interaction between gut microbiota and viruses seems to stabilize enteric viruses, enhance their adhesion to host cells, inhibit antibody production, and create a conducive environment for viral replication and immune evasion [125]. Conversely, several studies found that antibiotic-treated or germ-free animal models have higher rotavirus replication and more severe symptoms, which can be mitigated by FMT [124,126,127]. An elevated abundance of *Akkermansia* and a reduced abundance of *Faecalibacterium* were correlated with higher levels of viral antibodies [128]. Segmented filamentous bacteria, *Lactobacillus rhamnosus* and *Escherichia coli Nissle*, could protect against rotavirus infection, and *Enterobacter cloacae* could inhibit norovirus infection [8,9,129]. Van Winkle et al. reported that healthy microbiota could activate the interferon-λ (IFN-λ) pathway to safeguard intestines against viral infection [130]. IFN-λ pathway is necessary for viral resistance, for lacking IFN-λ could induce persistent infection [7]. A more complex condition was revealed that gut microbiota could inhibit norovirus infection in the proximal part of the small intestine but promote norovirus infection in the distal part of small intestine. The dual function was achieved by upregulating or downregulating IFN pathways in different parts of intestine [131]. Therefore, the function of gut microbiota on enteric viruses might be bidirectional under different conditions.

Enteric virus infection might alter gut microbiota composition. Disrupted gut microbiota was observed both in rotavirus- and norovirus-infected patients with lower richness and diversity [9,124,127,132,133]. The gut microbiota in rotavirus carriers were dominated by antibiotic-resistant Gram-negative bacteria [124]. The abundance of probiotics was extremely low and pathogenic bacteria were increased in rotavirus infected patients [124]. The *Bacteroidetes* and *Proteobacteria* increased while *Firmicutes* and *Bacteroidetes* decreased in rotavirus infected adults compared to healthy individuals [124]. Compared to healthy controls, *Escherichia/shigella* were both increased in animal or patients with rotavirus, and *Escherichia coli* were significantly increased in patients with norovirus [124,127,133]. 

Surprisingly, some studies found that viruses might exhibit probiotic-like effects in germ-free environments. Kernbauer et al. found that norovirus could restore gut microbiota homeostasis, repair the epithelial morphology, and enhance gut immune function in germ-free mice [134]. These findings suggest the complexity of the interaction between enteric viruses and gut microbiota. 

### 5.2. Gut Microbiota Modulation for Therapy and Prevention

Notably, the application of antibiotics in animals is aimed at removing gut microbiota, which could not be replicated in humans. Therefore, the protective effects observed in germ-free mice are not necessarily applicable to human situations. The protective effects of antibiotics against viral infections observed in germ-free animals do not necessarily imply that antibiotics can be used to treat patients with viral infections. 

Several studies have explored the potential of probiotics or FMT to modulate the gut microbiota and enhance resistance against enteric viruses. However, the effectiveness of gut modulation therapy in different studies varied. Some studies have found that probiotics can shorten the duration of enteric viral infections, release symptoms, and reduce the length of hospital stay in patients [135,136,137]. A meta-analysis of nineteen randomized controlled trials (n = 1624) indicated a lowered risk of rotavirus infection with probiotics treatment [138]. However, a review analyzing probiotics’ efficacy in rotavirus-infected patients revealed a mixed outcome, with half of the trials reporting relief and the other half showing no significant change [139]. Currently, evidence for FMT’s efficacy against enteric virus infections is lacking, as most studies are confined to animal models. 

The gut microbiota might modulate the immune response to viral vaccines, especially oral vaccines. Higher diversity of the gut microbiota was associated with weaker immunogenicity of oral viral vaccine [140]. Compared to the non-responders of the rotavirus vaccine, the abundance of *Clostridium cluster XI* and *Proteobacteria* were significantly increased in responders [141]. *Escherichia coli* were positively related to the immune response to vaccine [141]. Probiotics may promote the host immune response to enteric viral vaccines. Increased T cell response to the rotavirus vaccine was observed in pigs colonized with probiotics compared to the noncolonized group [142]. Supplement of *Lactobacillus rhamnosus* regulates immune pathways and stimulates Th1 cell response [143]. In summary, further research is needed to fully comprehend the potential of gut modulation therapy and protection in protecting against enteric viruses.

In summary, due to the unclear role of the gut microbiome in the infection of enteric viruses, there is no strong evidence to support the application of gut modulation therapies in the infection of enteric viruses. However, gut modulation therapies may help improve the host’s immune status and enhance its response to enteric viral vaccines.

## 6. *Salmonella enterica Serovar* Typhimurium

### 6.1. Interaction of Gut Microbiome and Salmonella enterica Serovar Typhimurium

*Salmonella enterica* subspecies *enterica* serovar Typhimurium, commonly known as *S*. Typhimurium, is an important causative agent of gastroenteritis and has been identified as the most common serovar causing invasive nontyphoidal *Salmonella* disease in several regions [144]. Meanwhile, it has been extensively studied as a model organism [145]. Therefore, we selected this serovar as a representative in investigating interactions between *Salmonella* and gut microbiota. 

*S*. Typhimurium typically induces self-limiting, nonbloody diarrhea. The complex interaction between *S*. Typhimurium and the gut microbiome is a topic of active investigation. Some researchers proposed that increased susceptibility to *S*. Typhimurium after antibiotic use was attributable to disruption of commensal gut microbiota [146]. Recent research showed that, in general, SCFAs could exert inhibitory effects on *S*. Typhimurium. Tsugawa et al. found that SCFAs could suppress *S*. Typhimurium by activating inflammasomes in macrophages [147]. Butyrate and propionate were found to be able to inhibit *S*. Typhimurium biofilm formation in vitro [148]. However, butyrate might also be utilized by *S*. Typhimurium and act as a factor enhancing its invasion, suggesting its complicated role in *S*. Typhimurium virulence [149]. 

Information about the changes in gut microbiota caused by *S*. Typhimurium is insufficient. In a gnotobiotic mouse model, it has been demonstrated that *S*. Typhimurium appears to exploit mucin-degrading commensal microbiota. Supplementation with mucin-degrading *Akkermansia muciniphila* during *S*. Typhimurium infection allowed *S*. Typhimurium to dominate the bacterial community [150]. However, a recent study in C57B6J mice yielded controversial results, in which *Akkermansia muciniphila* reduced susceptibility to *S*. Typhimurium [151]. These findings suggest that gut microbiota interactions with *S*. Typhimurium may vary significantly in different organisms, highlighting the need for further research on this topic in humans.

### 6.2. Gut Microbiota Modulation for Therapy and Prevention

The protective mechanisms offered by probiotics in *Salmonella* infection diseases range from inducing an immune response [152] to direct interactions with the pathogen [153]. Transmission electron microscopy showed close binding between *S*. Typhimurium pili and *Saccharomyces boulardii*, highlighting their potential for interference with the pathogen [154]. *Lactobacillus*, *Bifidobacterium*, and *Bacillus* have been tested for inhibitory effects against *S*. Typhimurium. Some strains of *Lactobacillus* [155,156] and *Bacillus* [157] demonstrated the ability to reduce inflammation and protect the gastrointestinal tract in animal models. It was also found that *L. plantarum* could mitigate the influence of *S*. Typhimurium on gut microbiota, reducing potentially harmful bacteria such as *Proteobacteia* [155]. Notably, Peng et al. have explored the use of genetically engineered probiotics to counteract *S*. Typhimurium. In their in vitro study, engineered *Lactobacillus casei* overproducing linoleic acid was found to significantly reduce *S*. Typhimurium biofilm formation, suppress its invasion, and reduce its virulence [158]. In summary, experiments have partially revealed the mechanism of microbiota-*Salmonella* interaction and probiotic-*Salmonella* interaction, but probiotics have not been generalized as a therapeutic approach toward *S*. Typhimurium infection due to lack of clinical evidence.

## 7. Other Infections

### 7.1. Pseudomonas aeruginosa

*Pseudomonas aeruginosa* is an opportunistic pathogen, and its infection of the gastrointestinal tract is more often a result of gut dysbiosis [159]. *P. aeruginosa* can frequently become resistant to various antibiotics, and hospitalized patients are somewhat susceptible to it [160]. Coexisting with *Candida albicans* might enhance its ability to colonization and dissemination since these two microbes could reciprocally enhance each other’s pathogenicity [161,162]. There have been successful examples of using probiotics and FMT to treat its infection in humans [163,164]. Janapatla et al. reported that a kind of prebiotic extracted from marine plants, fucoidans, is helpful in clearing *P. aeruginosa* from the GI tract of mice by inhibiting the release of some virulence factors and promoting the growth of *Bacteroides*, whose growth could be suppressed using antibiotic treatment [165].

### 7.2. Staphylococcus aureus

*Staphylococcus aureus* is a common bacterium colonizing human bodies. It is also a major cause of nosocomial infection. Its increasing antibiotic resistance has become a general concern, especially since the occurrence of methicillin-resistant *Staphylococcus aureus* (MRSA). One study revealed that MRSA infection was associated with decreased levels of SCFA-producing bacteria, but overall, the impact of *S. aureus* gastrointestinal infection on the gut microbiota is not yet clearly demonstrated [166]. There have been attempts to apply probiotics in *S. aureus* infection; however, most of them were conducted in vitro or on animal models. *Lactomodulin*, a peptide produced by *Lacticaseibacillus rhamnosus*, was found to have a bactericidal effect on MRSA in vitro, which might provide a new lead for antibacterial drug development [167]. Wei et al. gained successful results when using FMT to restore microbiota disruption caused by MRSA infection [168], although the sample size was small, and the effectiveness of this method needs further research to verify.

### 7.3. Candida albicans

*Candida albicans* is a commensal fungus that can also be pathogenic opportunistically. It may serve as a model for studying the relationship between gut microbiota and fungal infections. Situations that can foster *C. albicans* over-colonization include antibiotic use, immunodeficiency, preexisting gut dysbiosis, and perhaps overproduction of bile acid [169,170,171,172,173]. SCFAs can suppress the colonization of *C. albicans. C. albicans* infection accelerates gut dysbiosis, promoting the translocation of bacteria, prominently *P. aeruginosa*, in mouse models [174,175]. Possible methods of inhibiting *C. albicans* infection include FMT and probiotics, especially lactic acid bacteria [174,176,177,178,179]. *Lactobacillus rhamnosus* can antagonize the growth of *C. albicans* by competing for nutrient and adhesion sites [162]. However, most of these results were demonstrated in animal models; their clinical benefits still need to be verified.

Meanwhile, another point of view holds that when in a commensal state, *C. albicans* may exert regulatory effects on host metabolism [180,181]. More interestingly, Tso et al. presented a fascinating experiment in which they evolved *C. albicans* into a little-virulent and host-protective commensal fungus [182]. This may provide researchers with a new perspective in dealing with dysbiosis-related healthcare problems. 

### 7.4. Giardia duodenalis

Parasitic infections have also been found to be closely related to the gut microbiome. Among them, *Giardia duodenalis* is a parasite relatively well-studied. It is a common gastrointestinal protozoan, mainly prevalent in developing nations. Its infection is characterized by various gastrointestinal symptoms, such as diarrhea and abdominal pain [183]. Its consequence may be long-term, persisting even after parasite clearance [184]. One striking feature of *Giardia* infection is that the protozoan does not invade the surface of small intestinal cells, creating a constant interaction with the gut microbiota. A study in mice demonstrated that dysbiosis caused by *G. duodenalis* is characterized by increased aerobic microorganisms and decreased anaerobic microorganisms [185]. As for probiotic influence on *Giardia* infection, several strains of *Lactobacillus* have been shown to reduce the presence of *Giardia*, either by directly acting on the trophozoites or secreting inhibitory materials [186]. The studies mentioned above were almost all conducted with animal models. The role of gut microbiota in human Giardiasis is not yet clearly elucidated.

## 8. Conclusions

In an attempt to provide a comprehensive view of the topic of gut microbiome and infectious diseases, we have selected several diseases and reviewed their relevant literature. Considering the diversity of gastrointestinal infectious agents, it is quite difficult to include all kinds of pathogens. The pathogens included in this review are of high clinical importance and have been relatively extensively investigated from the perspective of gut microbiome. They may serve as representatives for understanding the complex interplay between pathogens and the human gut microbiome. 

While the influence of the gut microbiome on the diseases mentioned above can be observed (as summarized in Figure 1), during infection of different species, the crosstalk between pathogen and gut microbiome differs. Not unexpectedly, the application and effectiveness of gut modulation therapy vary to a great extent. From the perspective of therapeutical application, to summarize, evidence for the effectiveness of FMT in the treatment of *C. difficile* is most abundant among the diseases discussed above. Using probiotics for the prevention of *H. pyori* infection and cholera is a novel direction that may undergo appreciable development. Diagnosing cholera with engineered bacteria is also an interesting and innovative attempt. As for enteric viruses, despite the considerable amount of research, controversy has not yet been settled. A more in-depth investigation is needed to gain a clearer view of this question. What is more, even though alterations of the gut microbiome can be observed in various infectious diseases, little is known about the mechanism of these alterations, which also awaits exploration into more detail and may be meaningful for improving our understanding of the interaction between our gut microbiome and pathogens. 

It is also worthwhile to mention that although numerous studies have aimed to reveal bacterial change after infection, studies on the compositional alteration of gut fungi remain limited. Among the diseases covered in this article, data on fungi change are only found in *H. pyori* infection and CDI. Despite the fact that viruses were discovered to be able to directly integrate with gut fungi and some bacteria can synergistically or antagonistically interact with gut fungi [162], information about interactions between gut mycobiome and other infectious agents inside human bodies is still lacking. Gaining a more comprehensive view of the gut mycobiome, especially during disease states, may help renovate our knowledge bank in the field of gastrointestinal infectious diseases. 

## Figures and Tables

**Figure 1 ijms-24-15654-f001:**
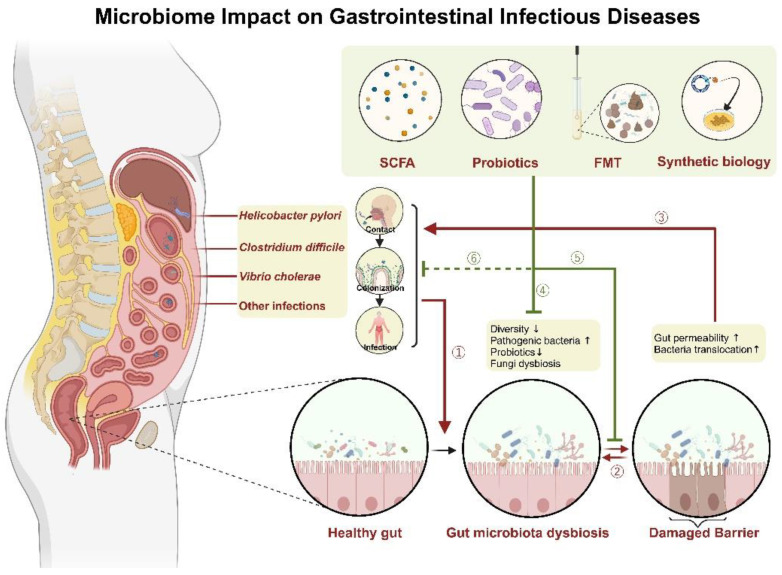
The Interaction Between Gut Microbiota and Gastrointestinal Infections. Created with BioRender.com, accessed on 26 August 2023. After entering the host’s gastrointestinal tract, the pathogen undergoes a colonization process, establishing infection. The colonization or infection by pathogens can induce dysbiosis in the gut microbiota (①), leading to damage to the gut barrier. Conversely, the disrupted gut barrier can also exacerbate dysbiosis in the gut microbiota (②) and create a favorable environment for pathogen infection (③). Gut modulation therapies, such as SCFA supplementation, probiotics, FMT, and synthetic biology interventions, have the potential to improve gut microbiota dysbiosis and facilitate the repair of the gut barrier (④,⑤). However, further evidence is needed to determine whether gut modulation therapies can directly inhibit pathogen proliferation (⑥).

**Table 1 ijms-24-15654-t001:** Fungi changes of different infections under untreated or treated situations in humans.

	Author	**Disease States** **vs. Control ***	**Microbiome** **Specimen**	**Fungi Change**
** *Helicobacter pylori* ** **untreated**	Suárez-Jaramillo [17]	37 infected vs. 38 uninfected	Proximal duodenum biopsy	No *Agaricomycetes*
	Dash [16]	12 infected vs. 48 uninfected	Fecal	*Candida glabrata* and other unclassified fungi ↑ **
** *Helicobacter pylori* ** **treated (triple therapy)**	Guillemard [29]	136 infected, triple therapy, before vs. after	Fecal	Fungi to bacteria Shannon ratio transient ↑ *Candida* transient ↑
** *Clostridium difficile* ** **untreated**	Sangster [31]	12 CDI vs. 12 non-CDI diarrhea	Fecal	*Penicillium* ↑
	Zuo [32]	34 CDI vs. 24 HC	Fecal	Diversity, evenness, and richness ↓ ***Ascomycota*** phylum ↑ *Candida albicans* ↑
	Stewart [33]	18 CDI vs. 31 non-CDI diarrhea	Fecal	*Aspergillus* and *Penicillium* ↑ *Oscillospira*, *Comamonadaceae*, *Microbacteriaceae*, and *Cytophagaceae* genus ↓
	Lamendella [34]	10 CDI vs. 10 non-CDI diarrhea	Fecal	***Ascomycota*** phylum, *Pleosporales* order, and *Dothideomycetes* class ↑ *Pichiaceae* family ↓
	Cao [35]	58 CDI vs. 91 non-CDI (28 asymptomatic carriers, 32 HC, and 31 non-CDI diarrhea)	Fecal	Diversity and richness ↓ ***Ascomycota*** phylum, *Pichia* genus, and *Suhomyces* genus ↑ *Basidiomycota* phylum ↓
	Cao [36]	58 CDI vs. 28 asymptomatic carrier	Fecal	Diversity and richness ↓ ***Ascomycota*** phylum ↑ *Basidiomycota* phylum, *Aspergillus*, and *Cladosporium* genus ↓
		58 CDI vs. 32 HC		
** *Clostridium difficile* ** **treated**	Zuo [32]	9 CDI-FMT Responder vs. 7 CDI-FMT non-Responder	Fecal	Richness and diversity ↑ *Saccharomyces*, *Aspergillus*, and *Penicillum* genus ↑ *Candida* genus and *Candida albicans* ↓

* CDI: *Clostridium difficile* infection; HC: healthy control; FMT: fecal microbiota transplantation. ** ↑: increase, ↓: decrease, same in Table 2.

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
