# Peer review of "Therapeutic Approach Targeting Gut Microbiome in Gastrointestinal Infectious Diseases"

_ijms, 2023, doi:10.3390/ijms242115654_

Round 1

Reviewer 1 Report

Comments and Suggestions for Authors

"The review titled “Gut microbiome: pathogenetic role and therapeutic approach in gastrointestinal infectious diseases” by Z. Han, Y. Min, et al. focuses on microbiota-related therapeutic approaches for treating various gastrointestinal infections. The recent probiotic and/or FMT treatments for these infections are summarized and discussed. While the topic is interesting, this review requires additional information and discussion before it can be accepted for publication. Therefore, the reviewer recommends a major revision."

Major:

1.     The title is confusing. The review does not primarily focus on the gut microbiome but on approaches to modulate specific gut microorganisms using FMT and probiotics.

2.     Please provide a clearer justification for the selection of pathogens discussed in this review. The discussion of "other infections" appears superficial.

3.     The conclusions for each subchapter are too broad and repetitive. Please summarize the main points and provide clear, concise conclusions.

4.     Figure 1 is overly complex. Consider simplifying it or adding additional descriptions to improve clarity.

5.     Clearly indicate when referring to studies conducted on animals or humans for better context.

6.     The findings about mycobiome changes, while interesting, do not seem applicable to the review's main topic. Please justify the inclusion of this table in the context of microbiota-related therapy.

Minor:

1.     In Table 2, the word "human" is not necessary here; consider moving it to the table title or description. Also, ensure that references are properly cited at the end of the table.

2.     Check for typos in the text, such as "Escherichia_shigella."

3.     Table 2 appears in the text before Table 1; consider reordering them for consistency.

4.     Revise confusing statements, such as: “Using probiotics for prevention of Hp infection and cholera, along with diagnosis of V. cholerae infection, is a novel direction of application that, though naive, may undergo appreciable development.”

Reviewer 2 Report

Comments and Suggestions for Authors

In the present manuscript, Han et al. discuss the impact of the gut microbiome on gut infections by Helicobacter pylori, Clostridium difficile, Vibrio cholerae,  enteric viruses, and other microorganisms. Pathogen interactions with the gut microbiome and how the latter could be used in the prevention and treatment of those infections are discussed. Controversies are also highlighted. It is an interesting study in its field.

Minor issues:

1. In the last paragraph of the introduction, please add references to the sentence: "For example, some researchers claim that the gut microbiota promotes infection of enteric viruses [REFs] while others claim it protective [REFs].".

2. Please confirm to follow standard rules for the identification of microbial species. While mentioning a species for the second time, please use the abbreviation of the genus and the full name of the species (such as "H. pylori"). In general, this is correctly done throughout the manuscript; however, there are some exceptions that should be corrected. Please check the repetition of "Clostridium difficile" (instead of "C. difficile") on page 5.

3. On Table 2, please type the bacterial species in italics.

4. In Figure 1, please check the italics in "Vibrio cholereae" (should be in italics) and "Other infection[s]" (should be plural, and not in italics). In the figure title, please also correct "(...) Gastrointestinal Infected Diseases" to "(...) Gastrointestinal Infectious Diseases".

5. Please carefully check the English quality of the manuscript. Sentences that should be improved include:

5.1. The sentence "Helicobacter pylori (Hp) is a pathogen that colonizes in the stomach of human and distributes worldwide" would be better as "Helicobacter pylori (Hp) is a pathogen that colonizes in the stomach of humans and is distributed worldwide".

5.2. The sentence "In the context of microbiome composition change, firstly and not surprisingly, in the gastric microbiome of Hp infectious patients, Hp becomes the dominant bacteria [15]." would be better as "In the context of microbiome composition changes, firstly and not surprisingly, in the gastric microbiome of Hp infectious patients, Hp becomes the dominant bacterium [15].".

5.3. The sentence "The specific phyla and genera of microorganisms changed differed between studies (...)" would be better as "The specific phyla and genera of microorganisms that were different among studies (...)". Please use "among" and not "between" when mentioning differences among more than two studies.

5.4. On page 11, please correct "Other infection" to "Other infections".

Comments on the Quality of English Language

Please see point 5 above.

Reviewer 3 Report

Comments and Suggestions for Authors

This is an interesting review on the role of the gut microbiota in treatment and prevention of different infectious diseases affecting the gastrointestinal tract. Overall, the review is well written although as the review unfolds, the writing loses its initial flow. There are number of suggestions for the authors that I hope will help to improve the review:

1- It is unclear why authors focused on those pathogens and no others. There are other more common infectious agents causing GI pathology such as Salmonella, or other important infections such as those caused by Giardia in which the microbiota could play a role, that are not even discussed.

2- Many observations discussed by the authors such as higher diversity in the context of H pylori infection, could be confounded by the medications taken by patients (e.g. PPIs). This needs to be highlighted and properly discussed.

3- Table 1 and 2. I encourage the authors to include numeric values (e.g. effect size, odds ratios etc) reported by the authors of those studies, so the reader can evaluate the importance of those variables in those specific outcomes.

Round 2

Reviewer 1 Report

Comments and Suggestions for Authors

No additional comments. Check typos

Reviewer 3 Report

Comments and Suggestions for Authors

The authors have reasonably addressed the comments I raised in the first round of review.